Application of domain-specific modeling in kinetography and bipedal humanoid robot control

Djukić Verislav 1 2
Oros Dragana 3 spawn@uns.ac.rs
http://orcid.org/0000-0002-2244-5698 Penčić Marko 3
Lu Zhenli 4
1 Faculty of Computer Engineering, Union University, Serbia , Belgrade , Serbia
2 Djukic Software GmbH , Nürnberg , Germany
3 Faculty of Technical Sciences, University of Novi Sad , Novi Sad , Serbia
4 School of Electrical Engineering and Automation, Suzhou University of Technology , Changshu , China
Givargis Tony
Electronic publication date: 2025 May 27
Publication date: 2025
Volume: 11
Electronic Location ID: e2864
Received 2024 Sep 12; Accepted 2025 Apr 7
Copyright: © 2025 Djukić et al.
Copyright year: 2025
Copyright holder: Djukić et al.
License: This is an open access article distributed under the terms of the Creative Commons Attribution License, which permits unrestricted use, distribution, reproduction and adaptation in any medium and for any purpose provided that it is properly attributed. For attribution, the original author(s), title, publication source (PeerJ Computer Science) and either DOI or URL of the article must be cited.
License URL: https://creativecommons.org/licenses/by/4.0/

Keywords: Domain-specific modeling (DSM), Domain-specific languages (DSLs), Kinetography, Labanotation, Code generators, Bipedal robot, Robotic foot

Funding: Ministry of Science, Technological Development and Innovation 451-03-137/2025-03/200156 Faculty of Technical Sciences, University of Novi Sad 01-50/295 This research has been funded by the Ministry of Science, Technological Development and Innovation (Contract No. 451-03-137/2025-03/200156) and the Faculty of Technical Sciences, University of Novi Sad through project “Scientific and Artistic Research Work of Researchers in Teaching and Associate Positions at the Faculty of Technical Sciences, University of Novi Sad 2025” (No. 01-50/295). The funders had no role in study design, data collection and analysis, decision to publish, or preparation of the manuscript.

==============================
The article presents a new approach in the development of software for bipedal humanoid robot controllers, based on the construction and application of graphic domain-specific languages (DSLs). The notations used to describe dance movements and gestures are typical examples of DSLs. With certain extensions, related to the description of foot topology, sensors and actuators, such DSLs are applicable for modeling dance movements that would be performed by a robot. The existing software development methodologies in robotics have a purely mechanistic approach to understanding and implementing robotic tasks. Such an approach in humanoid robotics complicates the understanding of the problem, as well as the specification and implementation of solutions. Our approach, which uses DSLs, adopts complex movements and gestures performed by the feet of dancers using professional dancers, people with above-average motor skills, as reference. We believe that the developed software can also be successfully applied to assistive robots that would help people with special needs whose mobility is significantly lower than average.

Introduction

Special-purpose humanoid robots require special types of controllers—software and hardware, the development of which typically requires several years of work. The most complex problem encountered in this process is the unpredictable space in which the robot performs movements and actions (Nguyen et al., 2022). This results in an unpredictable state space of control logic and user applications, which drastically complicates software development and testing (Papallas & Dogar, 2022). To overcome this problem, we use domain-specific modeling (DSM). In the creation of DSM language the specification of syntax and semantics incorporates expert knowledge in a way that includes visual and spatial representations of robot parts, its environment, actions, reactions, motions, etc. Such graphical domain-specific languages (DSLs) are deprived of any programming language or textual representation of language elements. The language concepts for expressing both static and dynamic features are handled by properties of language elements that are grouped in meta-concepts: objects, relations, roles, graphs and, optionally, ports. In this way, the state space of the control logic is drastically reduced to those states that are possible, more likely or based on usable heuristics and experience (Djukić & Popović, 2023; Djukić, Popović & Tolvanen, 2016). DSM is a pragmatic approach to the development of humanoid robots that simplifies understanding and specification of an “infinite” number of states. It uses a methodology that efficiently integrates expert knowledge in the field of robot application, with the expert knowledge of mechanical, electronic and software engineers. The time required for the development of bipedal humanoid robots and the final result also depend on the experience and organization of the development team. Language, i.e., the knowledge and understanding of the language of the robot application environment represents a tool that enables good organization. It is the language of experts from a specific profession rather than programming or general-purpose graphic languages that should be used for the formal specification of user requirements and features of robots (Kelly & Tolvanen, 2008). In practice, there is a gap between the domain language and the language used to specify robot features. That gap is even bigger for general-purpose programming languages, in which robot controllers are programmed (Jiménez et al., 2020).

This article presents an approach to developing formal specifications of a bipedal humanoid robot, its movement and actions, using graphic languages and models (Video: https://youtu.be/XlcLW5_TZNE). These specifications are automatically transformed, using a code generator, into a robot controller implementation (hardware) or just a controller software implementation. The approach is based on DSM, whose fundamental part are DSLs (Kelly & Tolvanen, 2008), as shown in Fig. 1. These languages are graphical—their construction is called meta-modeling, while the usage of such languages is called modeling (Kelly & Tolvanen, 2008). Each model is a single instance of the model type (graphical language) defined by the meta-model. The goal of modeling in humanoid robotics is to express the semantics of relationships between the given job, the properties of the robot and the tools that the robot uses to realize the given job (Djukić, Popović & Tolvanen, 2016) at a high level of abstraction. If necessary, each element of the model is decomposed into smaller units, up to the level that completely and accurately describes the topology, movements, states and actions. When it comes to movement and dance, a robot’s primary tool is its feet. However, the success in the development of that tool–the foot, as well as the entire humanoid robot, also depends on properly distributed roles among the experts participating in the software development (see Fig. 1).

Figure 1 Roles of experts in robotization.

When it comes to developing typical industrial robots, the roles of software, electronics, mechanical and automation engineers can be easily defined. There are decades of experience in these jobs. However, the assignment of roles and the determination of their frameworks become significantly more complicated when the team is expanded to incorporate experts in robot applications that cover non-industrial application domains. We cannot reduce the role of the domain expert—Fig. 1(2), to the role of an engineer from a production plant or a researcher from a laboratory. To solve this problem, we need to include dancers and choreographers as domain experts. They are users as well as participants in the construction and verification of a DSL for modeling mechanical properties, movements and actions of robots (Kelly & Tolvanen, 2008; Krahn, Rumpe & Völkel, 2006). If we burden them with general or incomprehensible technical language, we will lose them as invaluable consultants. Therefore, for bipedal humanoid robots, the use of the language of kinetography is both a compromise and a good solution. Another group of experts comprises engineers from the fields of mechanics, electronics and industrial standards, as well as robotics (Fig. 1(3)). Finally, the third group of experts are software engineers (Fig. 1(4)) whose role is in this case much more complex than in the development of industrial robots. Starting from modest software standards, they are expected to construct modeling languages, develop software tools and create a target system that executes the specifications given by the models. In order to meet the required standards for the reliable and safe operation of robot controllers, the modeling language is in some cases extended with elements (language constructs) for the precise specification of rules and procedures that apply the standards. However, the practical results are still modest, because ensuring reliability at only one level in the DSM hierarchy is not enough as reliability is required at each of the four levels. This is why our approach relies on software tools that allow fast reference implementation (Dantam & Stilman, 2012) and testing DSLs via model execution (Kelly & Tolvanen, 2008; Đukić et al., 2013). Model execution spans all levels in the DSM architecture.

The efficiency of the multidisciplinary development team with diverse expertise can be improved by choosing a suitable methodology—in this case, DSM and the DSM solution architecture with four levels (Fig. 1(1)). Iterative construction, application and refinement of DSLs ensures parallel development of the code generators, libraries (framework) and target system that executes the specifications, e.g., Robot Operating System (ROS). The methodology for software development or production, which is focused on the software development life cycle (SDLC), distinguishes several phases, such as decision, domain analysis, design, implementation, verification, deployment and maintenance. Our previous research and tool presentation (Djukić & Popović, 2023) describes a methodological comparison between our DSM-based approach and a typical SDLC approach. When it comes to software development for robot controllers and applications, we focus on the rapid language construction and its verification through a reference implementation. Our approach: (i) supports the semi-automated refinement of DSLs by means of modifiers, (ii) simplifies specification and implementation of DSLs by using advanced techniques for mapping language concepts to implementation concepts in the target environment for end-user applications and the environment in which the DSM tool works (.Net), (iii) generates source code automatically by means of code generators that are executed at the time of drawing the model and in run-time (model execution, when connected to real hardware), (iv) supports incremental deployment (update) in background, thanks to our own compiler and real-time system, without stopping the control logic, and (v) reduces maintenance of software versions to maintenance of model versions from which the source code is generated. In the process of the so-called conceptualization—recognition of linguistic concepts for modeling, it is necessary to make a certain deviation from the mechanistic view of robot movement and actions. The continuous flow of actions should not be perceived as a sequence of selected or random views, events, or images, aggregated in movement. The limited understanding of motion cannot be attributed to the limitations of either mechanics or electronics, but to the difficulty people have in understanding parallelism, continuity, space, and reaction to action. If we provide modeling languages with constructs for modeling space and time as sequences of images to convey the continuity, our approach is wrong. That is why even today, after several decades of robot development, we have humanoid robots whose movements are not human-like. The absence of spontaneity and naturalness largely stems from the shortcomings of the language used to program robots. It is not easy to express either parallelism or continuity using general-purpose programming languages. Robots’ “body language” indicates that their controllers do not understand the language of continuity, which is why robots cannot produce human-like movements. Those controllers indicate that the programming or modeling language used reduces the specification of the goal and purpose of an action to “trajectory language”. We see this very problem as the motivation in the software development methodology for bipedal humanoids, robots that would perform spontaneous, elegant movements and gestures during dance. Spontaneity, which inherently possesses individualism, cannot be modeled without incorporating the variations in the topological properties of feet, variations of actuators and sensors, as well as variations in the styles of interpretation of imagined goals, movements and actions. In the end, all this depends on the real environment in which dance is performed. We believe that modeling languages should first of all separate the goal of movement from the trajectory and provide some freedom in choosing the trajectory to the goal. If we look at the domain of space, a “freely chosen trajectory” is a resulting curve, which is created by complex modulations. Looking for the exact formula of that modulation is both complex and probably unnecessary. Instead, we focused our attention on movement tendencies and the so-called “algebra of shapes”, which in the domain of space is partially reduced to the “algebra of trajectories”. Seen in the time domain, the set of points that represent events, actions of the robot and reactions of the environment, and especially actions of the control logic towards actuators and reactions to sensors, is important. The mentioned algebras are the framework by means of which those points in the time domain are located both in advance and in runtime and determine the actions depending on the goal and purpose of the movement.

This article aims to present a new software development methodology for a controller that controls the leg and foot of a bipedal humanoid robot. Based on Labanotation (Fügedi, 2016), as a typical kinetographic DSL, the controller interprets parts of the traditional folk dance—Serbian kolo. The complexity and spontaneity of movements and gestures in such dances requires a control logic that is influenced by numerous parameters that are known only at the time of interpretation (run-time). This is an important difference compared to typical industrial robots. The presented methodology can be applied to solving problems related to numerous variations of foot construction, as well as styles of interpretation of the same models of movement and gestures, i.e., different styles of interpretation of the same notation (Djukić & Popović, 2023; Tolvanen & Kelly, 2019).

The article is structured as follows: “Introduction” describes the motivation and goal of the research; “State of the Art” analyzes the state-of-the-art, from the aspect of two groups of problems; “Problem Description” discusses in detail the issues related to this work; “Methodology”, based on the set requirements, presents the developed software in detail; “The Results” and “Discussion” presents the discussion of the results and the practical utility of the software tool through an example, while “Conclusion” concludes this article and provides directions for future research.

State of the art

Considering the aim of this article, the review of the available literature included two groups of problems: (i) Labanotation and (ii) DSL and their application in humanoids.

Labanotation

Labanotation is a graphic language—a form of notation, for describing traditional dances. As a language of kinetography, Labanotation uses a set of abstractions and symbols that describe movements, gestures and other elements of dance. We consider here the application of Labanotation in bipedal robots.

Music-driven dance system of humanoid robots, where the created algorithm divides the musical work into a series of phrases, is presented in Qin et al. (2018). In the given framework, music and dance have close connections in emotions and structure; dance phrases are selected from a pre-designed library using the “chance method”. The development of a framework called the Computational Laban Effort (CLE) system, for generating and characterizing expressive movements of people and humanoid robots, based on Laban Effort (LE), is shown in Simmons & Knight (2017). Using this system, it was determined that robot movement characteristics such as velocity, acceleration, range and timing are related to human LEs. The use of Kinetography Laban (KL) to generate motion in humanoid robots is presented in Salaris, Abe & Laumond (2017) where the use of Stack of Task (SoT) software generates dance movements for the humanoid robot Romeo. The movements are recorded using a motion capture system from which Labanotation is subsequently generated. A system for training robots in dancing is presented in Manfrè et al. (2017); the vision system registers the movements of the dancer, after which the acquisition system groups the movement sequences depending on the musical rhythm and beat. The developed framework allows the robot to learn dance movements just by observing the dancers. The possibility of applying Labanotation for programming humanoid robots using the SoT framework is shown in Salaris, Abe & Laumond (2016); the authors recorded a series of human movements needed to lift the ball from the floor, which they further implemented on the HRP-2 humanoid robot. This provided the basis for comparing the robotic and human movements. The development of a framework for adding a certain model of emotions to the predefined basic movements of humanoid robots, based on Laban motion, is shown in Cheng & Hsu (2015): the graphic interface tracks the music emotion locus in real time, on the basis of which the robot controller controls the movement of the humanoid robot with 17 degrees of freedom (DoFs). An aerial robot (drone) emotion expression model, based on Laban Movement Analysis (LMA), is presented in Gao et al. (2014); by mapping effort factors according to Laban’s theory, in the Pleasure-Arousal-Dominance (PAD) space of emotions, an emotion model based on the parameters of the drone’s path was obtained. A hierarchical three-layer motion programming architecture for humanoid robots that enables priority coordination during multitasking is presented in Han & Park (2013). The authors have developed: multitasking motion description language (MDLm), high-level motion primitives for an abstract humanoid for performing free space motions and tasks involving object interaction, and a new software architecture that facilitates the development of humanoid movement programs. Using Labanotation, an analysis of several steps from Taiwanese indigenous dance to develop a new dance technique and application that can be implemented in humanoid robots is presented in Hu et al. (2013).

DSLs and their application in humanoids

The development of two domain-specific languages—TaskDSL4Pepper and StateDSL4Pepper, for controlling the movement of the humanoid robot Pepper used for therapeutic purposes, is presented in Forbrig et al. (2023). TaskDSL4Pepper is based on task models, while StateDSL4Pepper is based on hierarchical state machines. The concept was driven by the idea that users—therapists, could program the robot themselves. Software for the rehabilitation of stroke patients is presented in Forbrig, Umlauft & Kühn (2023); a task-based language contains predefined robot tasks: commands such as raise your hand, turn around, etc., which are forwarded to the robot via Message Queuing Telemetry Transport (MQTT) messages; the proposed language is extended with the concept of sensor state as a prerequisite for the execution of tasks. A framework for trajectory optimization and an implicit hierarchical whole-body controller (IHWBC) for controlling the movement of biped robots are presented in Ahn et al. (2021); this open-source framework enables modularity because given modules can be replaced by other motion planners and whole-body controllers. A model-based design (MBD) software architecture for rapid prototype robotic controllers is presented in Ferigo et al. (2020); this open-source framework abstracts common robotic uses of middleware, optimizers, and simulators. Hardware and software design of a high-performance humanoid robot ARMAR-6, are presented in Asfour et al. (2019). The authors developed the ArmarX framework, which consists of three levels of abstraction—real-time control and hardware abstraction, perception and robot memory system, as well as task planning. DSL for model-driven scheduling of real-time tasks for robotics systems is presented in Wigand & Wrede (2019); the proposed language was integrated into the CoSiMA framework and tested on the humanoid robot COMAN. The software architecture, designed and implemented in RoboCup 2018, is presented in Martín-Rico et al. (2019). This architecture follows a three-layer organization, where the core represents a classic planner that uses the Planning Domain Definition Language (PPDL); the upper layer handles the symbolic domain, the middle layer calculates the sequence of actions to implement the goal, while the lower layer implements the actions that the robot should perform. This architecture implemented a human-machine communication system and a topological system for robot navigation around and over obstacles in the room. The methodologies for code generation, templating and metascripting for state machine assembly are presented in Ridge, Gaspar & Ude (2018). The authors developed an Application Programming Interface (API) for the modular development of programs to control the operation of robots that they tested in a robot simulation. The software architecture for human-guided autonomy implemented on the multilayer robot controller of the TROOPER DARPA Robotics Challenge team is presented in Gray et al. (2018); the autonomic layer consists of hardware and real-time services, the behavioural layer consists of simple actions, while the reasoning layer serves to initiate appropriate actions in accordance with the tasks. The software architecture of the iCub humanoid robot is presented in Natale et al. (2016) showing the way in which the robot abstraction layer was modified to respond to user requests and hardware changes. Software framework ControlIt! designed to support the development and study of Whole-Body Operational Space Control (WBOSC), is presented in Fok et al. (2016). The given framework enables fast instantiation and configuration of WBOSC parameters for practical applications such as the task of product disassembly accomplished using the Dreamer humanoid robot. A behaviour-based software architecture that decomposes the complex behaviour of a robot into actions—motor actuation, sensor detection, communication and system parameter setting, is presented in Martín, Aguero & Canas (2016); testing has proven that this architecture is suitable for robot soccer players and robots used in physical therapy for Alzheimer’s patients. A framework for multiple heterogeneous robots exploration using laser data and magnetic, angular rate, together with gravity (MARG) sensors is presented in Ktiri & Inaba (2013); given simultaneous localization and mapping (SLAM) framework can be applied to several types of robots such as drones, humanoid robots, etc. Also, the software enables the integration of map merging, where maps obtained from several robots are merged into a single global one. The interaction of a robot musician with a human partner through multimodal perception is shown in Pan, Kim & Suzuki (2010). The authors tested fundamental algorithms for initiative exchange on a robot prototype; using the vision system, the robot recognizes human movements and metallophone percussion, while based on sound analysis it recognizes the rhythm. A method that uses a BioVision Hierarchical (BVH) data file to automatically generate a dance score and print it in the form of Labanotation is presented in Chen et al. (2013); the authors obtained data on joint positions using the motion capture system; also, clock time is used as the basic unit for printing Labanotation. Task models for upper body movement operations of humanoid robots, based on Labanotation, are presented in Ikeuchi et al. (2018); bearing in mind that task execution is independent of the robot’s hardware, the same data collection module was used for all robots, while different modules were used for task mapping depending on the robot’s hardware. In addition, the authors determined that three different robots can automatically imitate the operations of the upper part of the human body with a satisfactory level of similarity, while a 3D sensor was used to detect the positions of the robot’s body parts. The combination of Labanotation with a system for programming robots—SoT, is presented in Abe et al. (2017); the authors analyzed the complex movements of humanoid robots and showed how dance scores are translated into robot movements using SoT; the obtained results were used to compare human and robot movements. The advantages of approaches based on different notation systems—Benesh and Eshkol-Wachman notations, Labanotation, keyframe animation and motion capture, as well as the combination of these approaches, are shown in Calvert (2016); to visualize the notation, the authors developed the LabanDancer prototype. The development of the MOVement-oriented animation Engine (MovEngine) computer application for creating animated movement, using a movement language based on the existing movement notation systems—Eshkol-Wachman Movement Notation (EWMN) and KL, is presented in Drewes (2016); it consists of a multipurpose library responsible for generating animated motion from notation-based instructions; in doing so, MovEngine can be a tool for creating and combining complex motion phrases. The analysis of human movements—as in Laban/Bartenieff movement studies (LBMS), in order to realize different movements (operations) of robots, is shown in LaViers et al. (2016). The result is the development of a software tool that allows dancers to be robot movement designers, as well as the translation between the different languages of LBMS and robotics; the ultimate goal is to apply tools to real robots and manage them via a web platform through a language that everyone can understand. The Benesh Movement Notation (BMN) system for analyzing and recording human movements is presented in Mirzabekiantz (2016). This system reduces body positions and movements in space over time to a series of two-dimensional key frames; BMN enables the identification of key frames in real time, as well as the display of the path and location of the body in space. In addition, the system analyzes the body scheme and focuses on movement dynamics. Two applications for recording and generating human body movements using Labanotation are presented in Choensawat, Nakamura & Hachimura (2016); LabanEditor application enables interactive graphic editing of Labanotation results and display of 3D computer-generated animation of score characters, while GenLaban application helps choreographers to create Labanotation score based on data obtained by motion recording.

Contributions

The analysis of the state-of-the-art research identified several solutions in the robot motion specification based on reverse engineering, i.e., recording a dancer’s movements using a 3D sensor. By recording the spatial position of body parts (e.g., joints) in relation to rhythm as a trigger for movement change, these solutions seek to extract movements and gests and translate them into a kinetographic notation—Labanotation. The disadvantage of these solutions is that since they are based on movement recording of this kind, they do not reliably distinguish between the elements that are necessary within the movement and those that are spontaneous or part of individual creativity. The authors mostly treat Labanotation as an isolated language of choreography, and robots as machines that only interpret kinetographic notations. The robot, as an interpreter, thus distances away from its original, i.e., the player or choreographer, whose movements are more creative and spontaneous. Also, the authors do not pay enough attention to the variations in the hardware of the robot (mechanics) and the dance interpretation. Finally, in the reviewed articles, it is not clear how the existing models are translated into executable specifications—the controller software, into run-time, which are required for the purposes of updating the control logic. In our opinion, generating good controller software is not possible using only the kinetography language, DSL must be extended as well. For this reason, it is necessary to establish semantic relations between models of movement and gestures (kinetographic notation) as well as feet and the effect of environmental forces and events, viewed in both temporal and spatial domains. The specificity of jobs which can be performed by bipedal robots indicates that it is necessary to develop and apply special “algebra” over numerous different semantic domains of properties. Having in mind the above considerations, we propose a new approach that enables the automatic generation of software for humanoid robot controllers with numerous variations of hardware (feet) and numerous variants of model interpretation.

Problem description

Humanoid robots do not perform movement and gestures spontaneously enough (Jiménez et al., 2020; Seyitoğlu & Ivanov, 2024; Macchini et al., 2022). In our opinion, the problem originates from the languages used for motion modeling or programming whose disadvantages are further transferred to the control logic of the controller. One of the approaches to solving this problem is based on the application of domain specific languages for the description of movement (Dantam & Stilman, 2012). A bipedal humanoid robot, which should interpret kinetographic dance notations, represents a complex system with a large number of different properties (Djukić & Popović, 2023). For these properties to integrate into a complex and functional whole, it is necessary to satisfy two prerequisites: (i) the completeness and precision of the specifications and (ii) the interpretation of each of them. The properties extend to areas such as mechanics, electronics and software. That is why, in solving problems, we prioritize the modeling of functional units over the separate modeling of software, electronics and mechanics. Such an approach ensures that we can analyze the properties of the robot before any part of it is actually made, which is essential. In the feet of bipedal humanoid dancer robots, we distinguish nine groups of control logic units, which are described using different types of models, such as (G1) topological properties of the foot, (G2) actuation and sensing for given topological properties, (G3) robot calibration, initial and in run-time, during the execution of the task, i.e., dance, (G4) description of movement and gestures or actions, (G5) description of the space as an environment in which the dance is performed, (G6) description of the forces acting in the real environment, which influence the expected movement (trajectories) and actions, (G7) mechanisms that provide feedback from the foot to the control logic of the controller, (G8) rules and procedures for ensuring reliability and safety of operation, as well as recovery to some valid intermediate state or initial state, and (G9) individual creations. From the nine groups listed above, G1, G3, G4 and G9 are further considered, while the others are not the subject of this article, mainly because they require the physical realization of the robot and an appropriate testing environment.

Topological properties of the foot

The human foot is a multi-segmented and flexible structure with 28 bones, interconnected by muscles and ligaments (Sekiguchi et al., 2020). A robotic foot that is “equivalent” to a human foot can differ significantly from the “original” in terms of its structure as well as in the number and shapes of the segments that it consists of. That is why the graphic language for modeling the structure and shape of the foot should enable the assignment of arbitrary topology, elements and relations between these elements, as well as the role of foot elements in such relations. The properties of the foot elements (joints, segments, tendons, etc.) are largely determined by their purpose—the function they should perform. In the case of footed living beings, those features are the result of evolution. The language for describing topological properties, like most languages for modeling real systems, consists of a part for describing static and a part for describing dynamic properties. Methodologically, this is correct. However, the problem arises if one ignores the fact that dynamic properties depend on static properties and vice versa. The shapes and the entire structure of the foot were created as a result of something that is dynamic—movement in certain conditions, with a certain goal. This fact must be recognized and expressed by modeling languages or languages for programming robot controllers. Reducing movement to the structure of a foot and to the trajectory is an approach that ignores both the purpose of the movement and the role of individual elements of the foot in this movement. Figure 2 shows the contours of several simplified shapes of different feet: (1–4) human foot without and with prosthetic replacements, (5) bird foot, (6) ungulate hoof, (7) forefoot of a human foot with prosthesis and (8) without prosthesis. Contours should be understood as curves or their sections, which are given analytically and which are connected to each other. Different transformations can be applied to them during movement, such as moving the start and destination points of the segment, rotation in space around the longitudinal axis, scaling, etc. A segment is not just a shape—it can be a sensor, an actuator or anything that initiates an action, reacts to a change in movement, or more closely determines relations to any other segment of the foot, trajectory or movement goal—Fig. 2. The segments can be split by selecting any point on the curve as a split point. If necessary, that point is moved in the run-time, e.g., the specification of the foot changes or the adaptation of the foot during movement is visualized. The operations on segments and the displayed representation greatly facilitate the introduction of new language concepts into the modeling language. Those new language concepts for modeling are a consequence of the expansion or refinement of the language, the essence of which is the growth of expert knowledge in a certain field of robot application.

Figure 2 Foot shape variations.

Although this approach has so far been tested separately on robot foot and hand, it is fully applicable to automatic software generation for an entire humanoid robot. Figure 3 shows a simplified model where all parts of the robot are described by curves that are used for a detailed description of movement, actions and environment. We call these curves “motion framework”. Essentially, they are different types of simple and complex curves, whose initial parameters can change dynamically. Movement, such as rotation of the shoulder joint, is modeled and displayed by spatial transformations of a subset of curves—rotations around the axis. The entire robot can be displayed in 3D and transformations, such as moving characteristic points, changing shape, scaling, rotations, etc., can be performed on it. Since all calculations are done relative to a joint or its support, whose position can change during execution, a robot with a complex topology can be parsed into: arms, legs, trunk and head. The existing number of libraries that make up our framework increases with each new foot topology. In the implementation of these libraries, we prefer to use simple algorithms to solve problems, compared to finding optimal movements.

Figure 3 A simplified robot model described by curves from motion framework.

Robot calibration

Bipedal robots need calibration during movement, but frequent stops would disturb the purpose of the robot dancer and its spontaneous movements. The control logic of the controller, both software and electronics, provides information about the discrepancy between the expected and the actual position through feedback, and most often through the encoder. If we take the trajectory as the only expected measure, the procedure of returning to the desired trajectory is quite simple. The question is whether this is the best solution is. Calibration should take into account the context of movement and multiple expected positions. A well-thought-out calibration procedure, for the purposes of interpretation of the kinetographic dance notation, can be translated into an elegant movement that does not leave the impression that the robot was in an uncalibrated state. Such more advanced calibration is focused on the range of error and the tendency of its changes, and not on the immediate reaction using a single signal to the robot actuator. By applying DSM and models, from which the software is automatically generated, it is possible to quickly test different calibration algorithms. They are automatically translated or embedded in the protocol for communication between the control logic of controllers, actuators and sensors. One model can describe several different calibration algorithms, which are not focused only on actuator signals, but on a higher level of abstraction, on movement and gesture, taking into account the topology of the foot.

Description of movement and gestures—actions

The description of movement and action can be realized using linguistic concepts similar to those used to describe the shape of the foot, regardless of whether the movement is oriented, while the segments of the foot do not have to be. We distinguish two cases of movement: (i) movement where the trajectory is at least partially known in advance and (ii) movement where only a part of the kinetographic notation and some recommendations for interpretation are known in advance. The graphic language for describing pre-known paths (trajectories) is more detailed, at a lower level of abstraction than the language that describes actions and places of their execution (Dantam & Stilman, 2012; Djukić, Popović & Tolvanen, 2016). On the other hand, the kinetography language is more abstract and suitable to provide the starting point or basis for the specification of software for robot controller. That language was derived from the knowledge possessed by domain experts, i.e., dancers and choreographers. A model described in one language can under certain conditions be translated into a semantic equivalent described using another language (M2M transformation). The basic prerequisite is the precisely defined semantics of both languages. Variations of views on the model are an essential property of modeling tools (Djukić & Popović, 2023). They are used when several different graphic representations of the same element types within the same language are needed. It is necessary to provide at least as many different views on movement as there are different roles within the team working on the development of the robot. In our solution, the description of the trajectories is synchronized with the kinetographic notation in run-time, using M2M transformations. That is why it is possible to change the place of performance, as well as the temporal and spatial sequence of actions and the way of their interpretation. Changing the reference points of robots and objects in space requires constant recalculation of movement. In a more complex case, this must be performed at the level of a millisecond. Information is acquired and exchanged via communication protocols used between controllers, sensors and actuators at such short intervals.

Graphic language for trajectories—Fig. 4, has both advantages and disadvantages compared to the kinetographic notation. The advantage is that, in addition to space, movements can also be roughly represented in time, so that a faster movement is illustrated by denser points, while a slower one is represented by less frequent points on the curve. This representation, or graphical syntax, has the disadvantage of illustrating gestures in one place. That is why, in various views of the movement model, some of the axes representing the spatial dimension are replaced by the time axis. In our approach, the problem is solved by the modeling tool which supports: (i) run-time transformations of the model, (ii) the construction and application of various representations depending on the state of the control logic originating from it, and (iii) through the continuous interaction of modeling tool and hardware, during which the controller is not given the entire trajectory, but goals or partial tasks. In this way, the “dance phrases” are noticed more quickly and the need for reverse engineering is eliminated.

Figure 4 Description of movement using curves.

Individual creations

The language of kinetography is a modeling language, and its application deliberately loses part of the information that is visible live or on video. Conjuring up the individual creativity of dancers and choreographers, or a robot that interprets choreographic notations, is at first glance only a matter of precise description of movements and gestures. From a technical point of view, that description that expresses individuality can be reduced to the calculation of the trajectory, elevation and rotation angles, under which the lower leg and foot approach a certain point in space, with a given speed and acceleration (Vukobratović & Borovac, 2004). That is still not enough, because the uniqueness of the interpretation is conditioned by the mechanical properties of the feet, shoes and the environment. All this is reflected in the specific features of the language that the dancer and the robot adopt. If a dancer or choreographer is recognized by some of their “dance language” or some of their “dance phrases”, then we have to keep that in mind. Also, we have to be prepared for a complete and rich modeling language to refer to only one single person. Is it rational to develop a graphic language to be used by only one person? Of course not. Is it rational to develop a methodology that effectively expresses specific expert knowledge and individuality that contains quality? In our opinion, this is justified and in fact, this is one of the basic goals of humanoid robotics.

The practical benefits of applying domain-specific modeling and DSLs for these purposes are multiple. Validation of the functional features of the robot is provided at the highest level of abstraction—the model level. Special language constructs serve to distinguish the goal from the trajectory, on entire sections of the movement or at least on some of them, for the sake of modeling spontaneous movement. Movements and gestures can be interpreted in different ways using the code generator, even for the same models. This makes it easier to experiment with different actuation algorithms. At the same time, more attention can be focused on movement tendencies, instead of focusing on a sequence of points in the spatial domain. Although the default interpretations of the model are useful, this is not what is ultimately expected in a highly specialized environment in which a robot is deployed. If some choreographers are satisfied with the default interpretation of the movement model, this means that the modeling language is semantically rich enough. However, this does not imply that that language will be able to meet the expectations of another dancer or choreographer (domain expert).

Methodology

To address the complexity, it is reasonable to examine separately different aspects of Labanotation interpretation by a bipedal robot: (i) movement, (ii) feet, (iii) environment, (iv) sensors, (v) actuators, etc. Nevertheless, a faithful interpretation is not possible if there are no formally described semantic relations between those elements. In general, a formal language is defined by abstract syntax, concrete syntax and semantics. The abstract syntax is usually defined as a meta-model or some grammar language—Backus normal form (BNF). Many meta-modeling languages do not describe the language completely. They implement a constraint language to describe static semantics, e.g., Unified Modeling Language (UML) is defined with Meta-Object Facility (MOF), plus with an Object Constraint Language (OCL)—the rule in OCL stating that inheritance among classes cannot form a cycle. Static semantics is used for model verification. When modeling languages are used, part of semantics is usually done via a generator, mapping the model to some other language (a programming language or some other modeling language) that executes specification. The abstract syntax and rules related to static semantics are defined in the meta-model and possibly in additional ways, e.g., in MetaEdit+ GOPPRR (Kelly & Tolvanen, 2008) meta-modeling language is used if possible, and otherwise MetaEdit+ Reporting Language (MERL)—code generator language. Code generation is not possible before the model is statically correct. As already mentioned, the rest of the semantics is defined by mapping to the target programming language, configuration file, etc. Semantic relationships have the knowledge that domain experts of different profiles exchange among themselves. Relying on domain experts is only one of the prerequisites for understanding and solving problems. Another very important prerequisite includes: the methodology and tools with which we formally describe the experts’ knowledge about the problems, their practical experience, and finally how quickly and easily we translate their knowledge into executable specifications (programs). In a typical scenario, software development phases (Fig. 5, stickers 6–11) are repeated periodically. Each of the phases is implemented by people with specific roles. However, regardless of the fact that this kind of job organization is logical and gives results, it has the following disadvantages: development, production, testing and maintenance of software are generally not done with integrated tools. In a well-designed DSM, multiple phases (6–11), most often 7, 8 and 9, are realized simultaneously using a much smaller number of roles and human resources. If the solution we are developing is based on the typical DSM architecture, where at the last level we have the target system—executor (run-time system, robot operating system), then the tasks from these two phases are executed simultaneously. The main question is how flexible the DSM tools are and what is the degree of integration of all the components that make up the DSM tools.

Figure 5 Software development life cycle vs. domain-specific modelling.

The productivity of developing software for the robot controller is the most efficient when the entire process of language construction and application (including deployment) takes place “on hot”. This means that the model is executed at the same time as the control logic generated from it. A model is a tool that synchronizes specifications, movements, actions, protocols, electronics and mechanics. Figure 6 will illustrate our approach—it contains three sub-models (SM_1, SM_2, SM_3) and the Labanotation of the dance that is mostly used by choreographers today (SM_4). The presented submodels represent the interpretation of the Labanotation.

Figure 6 A Labanotation interpretation model for bipedal robots.

The first submodel (SM_1), in the bottom left corner, with the annotation “Entry to dance notation” is a notation of an ethno dance1 using DSL for Labanotation. Compared to the notation of the same purpose that is shown in the bottom section of the picture (SM_4), this notation of ours is a model. It is based on a precise definition of the language for modeling and numerous layout variations of language concepts. Its basic elements are Bit and Measure object types. The properties are defined above the Bit type object determining the direction, height, level, gestures, accents, foot hook, etc. By establishing relations between arbitrary objects, the semantic links are redintegrated between those elements of the cinematic notation or any other elements of the model, which do not have to be in sequence as in SM_4. They are connected as preparatory actions that need to be done in order to successfully perform a movement that is more distant in time or space. However, this is not only about preparing dancers but about preparing robot resources for a certain action, those resources being time, space, energy, materials (props), etc.

Above the language and models defined in this way, a model query language was created for model searching and updating. The query language is not limited to language concepts used for Labanotation, movements or robots. One of the applications of this query language is finding similar and specific dance sequences and updating them. In the example, similar sequences are recognized, grouped into measures and marked with different light-shaded background colours. Queries about the model include its variations, submodels, etc. The interpretation of notations is done by code generators, for different needs. The code generator interpreter, as a tool that is part of the DSM architecture, has been extended to be able to modify the model. The generators that have commands to change the state of the model are called action reports (Đukić et al., 2013). During the interpretation of notations with SM_1, it is possible to make changes to any element (insert, remove, connect, disconnect), as well as change the value of the Bit properties (V_1). Semantic errors are prevented by the use of formal language definition and the interpretation of structural and value constraints. These constraints contain the choreographer’s knowledge that is not expressed through specific modeling concepts so that the language does not become too complex. Generators are programs whose algorithms are aimed at traversing through the graph. In practice, a dozen generators are needed to generate control logic, user applications and simulation. Here we describe two, whose purpose is visible in the shown model. The first, based on Labanotation, generates a voice/sound interpretation of movements, which to a considerable extent helps dancers, choreographers and robots to recognize commands without observing or interpreting models (images). The speaker symbol (1_2) shows which Bit (model element) is currently being interpreted. In this example for the tonal interpretation of Labanotation, the generator GenerSoundForStep is called from the main generator (GenerateDance Sounds) to process the first object (Step1). Then the graph is traversed by recursively calling the generator GenerSoundForStep. The algorithm checks the direction of movement of the left and right leg. Depending on whether the movement is performed by the left, right, both at the same time or by neither leg, the tones “d”, “f”, “c” or “g” are emitted. The duration of the tone depends on the height of the control (:Height as … as sound;), the duration can be specified by another property. Property values can be modified with more than two hundred functions (operators) that are optionally applied successively in the format “as op1 as op2 …”.

REPORT GenerateDanceSounds

var $par = ‘Step1’

GenerSoundForStep($par)

ENDREPORT

REPORT GenerSoundForStep

param $inBit = ’’

do :$inBit as objects;

{

   if :DirLeft; != ‘Unknown’ and :DirRight; = ‘Unknown’ then

       :Height as plus_d as sound;

   elseif :DirRight; != ‘Unknown’ and :DirLeft; = ‘Unknown’ then

       :Height as plus_f as sound;

   elseif :DirRight; = ‘Unknown’ and :DirLeft; = ‘Unknown’ then

      :Height as plus_c as sound;

   else

       :Height as plus_g as sound;

   endif

   wait 50

   do ~PrevStep>StepFlow~NextStep.Bit;

   {

       var $next = :id;

       GenerSoundForStep($next)

   }

}

ENDREPORT

Finding related objects through relations and roles of a certain type is achieved by expressions of the form:

do ~PrevStep>StepFlow~NextStep.Bit; { … }

This expression is interpreted as a request to find a collection of Bit type objects which, starting from the current object, can be reached through roles of type ~PrevStep, relations of type > StepFlow, and finally through roles of type ~NextStep.

The second generator serves to translate Labanotation as a dance model into an analytical description of movements for legs, arms and other body parts, which is shown on the submodel in the upper left corner (SM_2, Generated movements and gestures annotation). Movement is not only a trajectory but also a series of specifications in different formats related to the trajectory and to the tools (feet), at any part of the trajectory or at any point. When generating curves for the description of movement, the order of interpretation is not determined on the basis of the spatial arrangement of elements, but on the basis of given relations and roles in relations (connections). Therefore, all bits are connected, and the connection from the last element of the first vertical goes to the first element of the second vertical.

The submodel that occupies the largest part of the image (SM_3) serves to specify the topological and dynamic properties of the robotic foot. The topological properties of the foot are described directly, using the same set of curves (shapes) used to describe movement. Patterns are also used, in order to describe the feet more quickly. The basic types of curves that make up the framework for shape description are shown on the setLineType switch (3_1). In order to send information about the change of curve type to the robot controller, at the time of execution of the model, the event handler SendPropValueToRTS(PropVal) is defined above this switch. We distinguish the description of static properties from the description of dynamic properties, but these properties are mutually dependent. Static properties are described using curves that determine the contours of the section, the contact surface or the elements that make up the foot (joints, bones, tendons). Each curve, its segment or two arbitrary points on the curve potentially defines a zone over which some of the following transformations are performed: (i) segment removal, (ii) insertion, (iii) replacement, (iv) curve type change, (v) rotations segments in space, (vi) scaling by one of the dimensions, (vii) modulation, (viii) changing the domain or value of any property, etc. The slider (3_6) is used to set a current curve over which changes are made. The scales (3_2) are used to change the angle at which the 3D representation of the given foot topology is viewed. The dynamic properties of the foot are given by first adding a property of certain semantics to a curve or its segment. In simpler cases, this would be, for example, the maximum angle of rotation of the segment around the axis connecting their beginning and end. In more complex cases, these are properties used to describe tendons or muscles. With the help of submodel SM_3, a set of topologies of the robotic foot with which we want to experiment is obtained (3_5). A set of feet that was created by varying the input parameters is shown above the annotation “Foot variations”.

In the modeling of movement and gestures for arbitrary static and dynamic properties of the foot, there are two equally complex problems, which we solve using the “algebra of curves” or “algebra of shapes”. These are: (i) prediction and calculation of action and reaction, to correct the expected based on the realized, and (ii) visual representation of motions, gestures, robots and space. The first problem is solved by applying optimized calculations that work in real-time, based on the geometry of space. As for the display, there are possible compromises by replacing the complex high-resolution 3D presentation with a simpler 2D display that focuses only on certain properties. Algorithms for the calculation of movement, as well as those for the presentation, use property values that are transferred between model objects via roles (connections). This was a way of realizing “graphical programming”. In the image with the foot (3_3), all roles have property names that are exchanged. On the right side of the foot (3_3), in the form of a label, are the values of some transferred and some calculated properties. The actual source of parameters for foot actions is the submodel SM_2, which also has its own more compact representation that is the so-called “shallow copy”, object pathGen (3_7). The slider on the left is used to scale the given movement path, e.g., for simulations.

The results

Several tests were conducted to test the functionality of the proposed software. The functionality of the communication protocol and software execution was tested first, followed by the verification of the modeling language.

Communication protocol and software execution tests

The proposed software is executed in real time (using Serial Peripheral Interface—SPI, full duplex), which implies that all changes to the model are simultaneously propagated to the robot in real time; implementation on the robot solely depends on the robot hardware–the type of communication between the driver and the robot controller, the characteristics of the actuator, etc. Feedback information is obtained from: actuator encoder, current load measurement chip and force sensor.

In order to analyze the usability of the controller, we performed three tests. In the first test, we wanted to determine whether the software, definitions of motion, actions and environment can provide spontaneous elegant movements and programming motion by manually moving the robot’s foot. In the second test, we wanted to determine whether, as well as how quickly and reliably the predicted movements and the reactions to unpredictable situations can be changed. In the third test, we analyzed the best feedback cases for motion calculations.

Test_1: For a 4 DOFs foot with a load capacity of 1.5 kg, driven by stepper motors Nema 17-23, it is possible to perform movements that are three times faster than those realistically required in practice for walking or slower dances. Extremely fast gestures (operations) have to be realized using a smaller number of joints, usually one. The definition of movement is analytical, but at each point of the curve, gestures can be defined for each special mechanics (the foot in this case). On the initial description of the movement using motion frameworks, as well as the modulations that change that initial movement, precise analyses can be performed such as (i) length of the path; (ii) length of segments; (iii) percentage of the path traveled at certain points; (iv) analysis of time and necessary energy; (v) space for performing movement and action; (vi) deviations of the actual length of the path from the ideal; (vii) expressed analytically; (viii) spatial relationships between points at any point of the path; (ix) partial analyzes of the properties of movement on individual segments (sections) and (x) heavier points (spatial), (Fig. 7).

Figure 7 Calculations, predictions and assessment of movement space.

Test_2: The motion definition is a variable part of the control logic, independent of the source code. This definition, expressed using elemental or composite curves, can be changed in real time based on the data obtained from the feedback loop. The expected and the realized parameters are compared at the level of one movement pulse thanks to a protocol based on SPI communication which is also full duplex. The quality of feedback is determined by the driver hardware. For testing, we used a Trinamic chip (TMC262) because it can be used to detect changes in current, which enables recognition of the effect of gravity, encountering an obstacle, etc.

Test_3: Feedback, in addition to extremely high precision and reliability, also ensures the application of calibration algorithms in run-time. Calibration methods are different and one solution is not applicable to all robots. Nevertheless, the feedback loop with the motor encoders is a very good solution. Due to the fact that every movement of the given resolution is calculated during motion, it is not necessary to use a servo control.

The verification of the modeling language

The software tools we used to achieve fast and reliable verification of the modeling language are integrated into one unit. Reference implementation of DSLs, refinement and verification is simplified since it is possible to define a meta-model (language), draw models and write code generators in one place. These code generators are immediately tested. In the beginning, the model is empty, and the definition of DSLs is incomplete. Each subsequent refinement of each component is immediately tested on the target platform–run-time system, real-time operating system (RTOS). For the highest level in the DSM architecture, they can use MetaEdit Modeler and DVMExIDE (commercial tools). The second level, code generators, use MERL program interpreters. For the .Net environment, the syntax of the MERL language and its interpreter have been significantly expanded. At the third level, we use our own libraries for various types of movement and action calculations, protocols and communication through various hardware interfaces. The code generators generate programmable logic controller (PLC) and C++ code and configuration parameters, while our own compiler generates machine code for different processors and operating systems (when used). The performance of the generated code is at the level of optimized C++ compilers. The motion descriptions consist of a set of predefined shapes, whose number and parameters can be changed at run-time. The shapes can be the result of dynamic modulations, which gives the control logic the possibility to translate the influence of external forces into an analytical record of movements and actions in run-time.

The simplest scenario for generating and testing robot software consists of several parts (models) which, in a simplified form, are shown in Fig. 8.

Figure 8 Models of topology, movements, and protocols for controllers.

Figure 8 contains three models, but the model of topology is depicted within the model for motions. The link between them is the glRJRotMix element (Sticker 1). In the left, middle, part of the figure (Sticker 2), an initial set of curves describing the movement, is given. It can be a description of the foot movement of a dancer dancing the Serbian “kolo”. This description consists of a set of different types of simple and complex curves, such as lines, Bezier curves, sinusoids, pulses and circular slices. If we use DSL for Labanotation, then each of these strokes also contains properties related to speed, acceleration and other parameters that additionally describe a movement. Based on the given foot topology in this example (Sticker 3), a description is generated, which serves as an input to the Action function block (Sticker 4). Elementary movements (shapes) are connected to the state machine (Sticker 5), whose states change when a set of conditions is met. The state machine provides forward and backward movement, according to the conditions and applications in run-time. In the simplest case, one state corresponds to the definition of one movement. The parameters of the current shape are passed to the model elements in the upper left corner (Sticker 6). These include the starting and destination coordinates in space, number of impulses, amplitude, rotation around the longitudinal axis, etc. The PointProvider function block (Sticker 7) has the task of finding points belonging to the current shape’s path, taking into account the specified precision, time offset, acceleration, etc. This function block allows dynamic change of all input parameters, which means that the reactions to changes in the environment will be at the level of movement (one micron or 0.2 ms in the case of slower processors). When generating a code, object types, relations and roles are analyzed. In principle, PLC or C++ code is generated from one model for the roles shown in black, and an animation code (default application) for the roles shown in orange. However, the semantics and layout of the model, and its interpretation, is the decision of whoever constructs the DSL.

When the point, the movement, its properties, the environment in which the movement is performed, as well as the topology of the foot (which can also change dynamically) are known, the function block Action (Sticker 4) takes over these parameters and determines the angles of rotation and elevation of the segments in relation to the joints for the given topology. This is a typical problem in inverse kinematics. Each of the calculated angles should be translated into commands sent to the actuator, taking into account both the actual and the desired state, thanks to the feedback from the encoder. The actuator communication protocols can also contain numerous other parameters that define obstacle sensitivity, load, maximum speed, etc. The actuation method via the stepper motor is shown in the bottom part of the picture (Sticker 8). This part of the model practically receives the calculated values that need to be translated into signals to the actuator, which are sent via the protocol (Sticker 9). For testing, we used our own protocols via the SPI interface, full duplex, with speeds of 2 to 16 MHz. The performed tests determined that packet exchange at a speed of 2 MHz is sufficient to perform elegant leg and arm movements. The processor that performs motion calculations is embedded, at 2 GHz. The bottom part of Fig. 8 shows the elements that ensure reliable movement and its interruption, if necessary (Stickers 10, 11 and 12).

Discussion

The greatest practical benefit of constructing and implementing kinetography DSLs and DSMs in the manner shown in Fig. 6, is the automatic generation of software for predictive logic controllers. At the same time, predictive logic takes into account (i) unforeseen events in the environment that change the trajectory, (ii) deformations or changes in foot topology, (iii) changes in the space of movement and (iv) arbitrary movement problems. Figure 9 shows a part of the foot trajectory obtained from Labanotation. For such a movement, the length of the trajectory, the current position, the space for all or part of the movement, the type of movement, etc., are calculated in advance. Zones are marked on the trajectory where some events (E1 to E4) change the planned trajectory or change any properties that require a change in the foot control logic. The movement is not limited to a trajectory that is independent of the tool (feet). It takes into account reactions to events, which are modeled through changes in the properties of the foot. All the zones marked with different colours are also potentially zones of application of the “algebra of curves” in the run-time. That algebra works on properties whose semantic domains are different, and time, space, shape and speed are the most common domains. In the bottom part of Fig. 9, a possible way of processing event E1 is shown, which is realized by dynamic modulation of a circular segment (red segment with an arrow) using a pattern, a sinusoid with three half periods. Due to the fact that the initial trajectory is a circular slice, the height of the middle half-period is shifted towards the top of the slice. The solution with the algebra of curves is analytical and represents an efficient mechanism for recognizing elementary movements and gestures (phrases) and creating composite ones. Each modulation potentially becomes a special kind of gesture and style of dance interpretation, which is linked to a certain choreography, dancer or choreographer, or to robot with certain properties of the feet. Due to the possibility of reducing unknown and unexpected movements to an analytical description, this approach is applicable in machine learning, where reliable conclusions would be derived from a small sample.

Figure 9 Prediction of path, time required and modulation.

Our approach enables efficient testing of the movements of bipedal robots of different topologies due to the ability to refine the modeling language at the time of model execution. Essentially, this approach involves a fast reference implementation of the language, which proves its usability for modeling. It is aimed at generating controller software from models and during model execution, rather than at (solely) testing controllers via simulators. From the point of view of an original contribution to the development of software for robot controllers, we should highlight our approach to understanding and modeling movements. The logic of robot controllers, which is based on the forced movement of the feet along the points of the trajectory, is not sufficient, regardless of the fact that it is generally accepted in robotics. Instead, we focused on parts (zones) of movement that are bordered by events, which can be determined in time, space or any other domain. Those zones are potentially the ones that require control logic based on monitoring tendencies and experience, not just trajectories, thus providing the basis of predictive logic for controllers. In other words, those zones require attention to be focused more on continuity and the goal, instead of focusing only on the position at a certain moment.

DSM follows a typical top-down approach, solving problems by moving from higher to lower levels of abstraction. It does not reject the results that can be obtained by reverse engineering for the purposes of creating a framework (library). However, these are things of a lower level of abstraction. This is not a matter of the principles on which DSM is based, but rather a matter of experience that without understanding the language of movement and gestures one cannot create DSL for context-dependent actuation or calibration of bipedal robots. The control logic that we have described is not only focused on the electronics that drive the actuators but on the logic of action and reaction as well as on their relations to the numerous elements that affect the goal, not just the form of a movement. In the end, achieving spontaneity of movement is not possible if the properties of the tool with which it is performed (feet) and the properties of the available time, space and forces acting in the environment are not considered at the same time. A methodological approach that enables simple variations of the model to generate software for robot controllers and is adapted to the individual significantly shortens the time required for the implementation and validation of the controller.

Starting from the solution for conceptual modeling provided by the DSM tool MetaEdit+ Workbench (MetaEdit+ v5.0, 2024), and in order to verify the presented software development scenario for bipedal robot controllers in practice, it was necessary to expand almost all components in the architecture of solutions previously used for industrial robots (Djukić & Popović, 2023). It should be noted that an original software solution was developed (DVMEx IDE, 2024), whose advantages are: (i) the editor for the construction and application of DSLs is extended with graphic controls that have over 200 properties; including those with advanced 3D graphics that visualize the “algebra of shapes”, (ii) the code generator language—reports and its interpreter have been extended with new commands: for changing the model in run-time and for visual tracing (debugging), (iii) the target framework is supplemented with functions for the “algebra of shapes” and calculations for different topologies of the foot, (iv) the last level—the level of the executor (ROS, run-time system), is expanded with metalogic, commands for dynamically changing communication protocols and services for automatic diagnostics, time domain scaling, etc. This is in sharp contrast to most authors in the field who deal exclusively with individual components of the system when solving a problem (motion analysis using existing software, development of various frameworks and software for robot controllers), and partially with conceptual modeling, without a more advanced graphical representation of such language concepts.

Conclusion

A software tool for collecting and specifying a choreographer’s expert knowledge was developed by constructing DSLs of kinetography and expanding them with languages for describing feet and space. By using this tool, i.e., by drawing a model, different cases of movement, foot properties and surrounding space are described. Consequently, the existing editor for the construction and application of DSL has been expanded. Also, new commands for changing the model during its execution have been implemented. The target framework is supplemented with the necessary functions, while the target system (executor) is expanded with metalogic and appropriate commands and services. This methodology comes to the fore in robotics, both in terms of productivity and software quality, if each of the levels in the architecture is supported in the right way: (i) meta-modeling with possibilities for quick specification of graphic languages, (ii) generators with simple syntax and quick transformation of models into source code and other artefacts, (iii) framework (libraries) with functions that perform movement calculations for different foot topologies, variable geometries, etc., and (iv) a target system for which an advanced compiler creates binary code that provides various strategies for avoiding errors and restoring the control logic to a working state without external intervention. Since these features are not present in the existing General Public Licence (GPL) compilers, we developed our own compiler for Intel and ARM processor architectures starting from the PLC software and features of the robot control system. As a result, the developed software tool represents an advanced solution that can be applied in various fields, such as industrial and non-industrial robotics, construction, painting, medicine, etc. Future research will be focused on the improvement of the DSM tool in two directions: (i) development and implementation of a framework for “algebra of shapes”, and (ii) expanding the knowledge base about different protocols for communication with actuators and sensors, depending on the precision, complexity and reliability of the task that the robots need to perform.

Supplemental Information

Supplemental Information 1 Source for animation, metamodel and model example.

The .dvMod file contains a metamodel (definition of graphical domain specific language that is constructed for modeling in robotics), and one model example, that is described in the article.

The DVMod file is in XML format and some attribute values that are images are coded by Base64(), This is wide used coding for images and sounds in applications.

Other contents include: script code for animations, model execution, that illustrates how model is used for verification of language and models, before real hardware is done.

Supplemental Information 2 Domain-specific modeling in kinetography and bipedal humanoid robot control.

We owe a great deal of gratitude to the Cultural and Artistic Association “Svetozar Marković” from Novi Sad, Serbia, and their choreographer, Slavisa Đukić, for helping us understand the language of kinetography.

Nomenclature

DSL Domain-specific language

WBOSC Whole-Body Operational Space Control

DSM domain-specific modelling

MARG magnetic, angular rate, and gravity

ROS Robot Operating System

SLAM simultaneous localization and mapping

SDLC software development life cycle

BVH BioVision Hierarchical

CLE Computational Laban Effort

MovEngine MOVement-oriented animation Engine

LE Laban Effort

EWMN Eshkol-Wachman Movement Notation

KL Kinetography Laban

LBMS Laban/Bartenieff movements studies

SoT Stack of Task

BMN Benesh Movement Notation

DoF degree of freedom

BNF Backus normal form

LMA Laban Movement Analysis

UML Unified Modeling Language

PAD Pleasure-Arousal-Dominance

MOF Meta-Object Facility

MDLm multitasking motion description language

OCL Object Constraint Language

MQTT Message Queuing Telemetry Transport

MERL MetaEdit+ Reporting Language

IHWBC implicit hierarchical whole-body controller

SPI Serial Peripheral Interface

MBD model-based design

RTOS real-time operating system

PPDL Planning Domain Definition Language

PLC programmable logic controller

API Application Programming Interface

GPL General Public Licence

Additional Information and Declarations

Competing Interests

Verislav Djukić is the owner of company Djukic Software GmbH—Germany, and also employed at Faculty of Computer Engineering, Union University, Belgrade, Serbia. The authors declare that they have no competing interests.

Author Contributions

Verislav Djukić performed the experiments, analyzed the data, performed the computation work, prepared figures and/or tables, conceptualization, methodology, software, formal analysis, investigation, writing and original draft preparation, and approved the final draft.

Dragana Oros conceived and designed the experiments, performed the experiments, analyzed the data, authored or reviewed drafts of the article, conceptualization, methodology, investigation, validation, writing—original draft preparation, review and editing, and approved the final draft.

Marko Penčić conceived and designed the experiments, performed the experiments, analyzed the data, prepared figures and/or tables, conceptualization, methodology, investigation, validation, writing—original draft preparation, review and editing, and approved the final draft.

Zhenli Lu performed the experiments, analyzed the data, authored or reviewed drafts of the article, conceptualization, methodology, investigation, validation, writing—original draft preparation, review and editing, and approved the final draft.

Data Availability

The following information was supplied regarding data availability:

Code is available in the Supplemental Files.

1 The presented Labanotation is a notation of the sequence that is very common in Serbian folk dances, such as “Three steps in position–then jump forward”.

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
