# Peer review of "Application of domain-specific modeling in kinetography and bipedal humanoid robot control"

_PeerJ Computer Science, doi:10.7717/peerj-cs.2864_

## Round 0.1 · original submission · Major Revisions

The reviewers have provided considerable commentary and feedback on your manuscript. The paper presents an interesting approach, but it requires significant revisions to address these limitations and improve clarity, experimental validation, and broader applicability.

We encourage you to take a look at all the reviewer feedback, including these high-level take aways:

Limited Scope: The paper focuses primarily on foot movements and the topological properties of the foot, which limits the broader applicability of the methodology to full-body robot movement. The suggestion is to extend the approach to encompass whole-body movement.

Clarity and Accessibility: The explanation of the software framework and its components is difficult to follow due to the reliance on domain-specific knowledge of Labanotation and graphic modeling. A more accessible, step-by-step breakdown with examples would enhance readability.

Experimental Setup and Validation: The paper should provide more details on the experimental setup, test cases, results, and validation criteria for the software tool. Additionally, the tests should cover various scenarios and edge cases.
Related Work: The paper misses references to recent advancements in related fields, such as multimodal strain sensing or predictive modeling, which could provide valuable context for the research.

Reviewer 1 ·

Basic reporting

This paper presents an interesting approach to robotic movement control, using domain-specific languages (DSLs) inspired by kinetography, specifically Labanotation, to potentially enable bipedal robots to perform human-like dance movements. The authors introduce a methodology that departs from traditional mechanistic approaches by leveraging graphic-based DSLs to model and generate control software. The use of domain-specific modeling (DSM) offers a structured and iterative way to model complex movements at abstraction levels while generating executable code for real-time application.

However, while the work demonstrates potential, there are some limitations/suggestions to further improve:
1. A primary limitation of the current work is its exclusive focus on foot movements and topological properties of the foot. This narrow focus limits the broader applicability of the methodology to whole-body movement. Discussing the potential solutions to extend this method to the robot’s whole-body movement could add value.

2. The methodology depends heavily on specific hardware configurations, particularly the construction of the robot’s feet. Differences in foot mechanics and hardware limitations could challenge the usability of the proposed approach. How the foot morphology used in this work transfers to the robot’s foot is unaddressed in this work.

3. The paper's explanation of the software framework and its components could be clearer and more accessible. The current description requires substantial domain-specific knowledge of Labanotation and advanced concepts in graphic modeling, making it challenging for readers unfamiliar with this field to follow. Including a detailed breakdown and step-by-step explanation of each software element, along with practical examples, would improve the readability and accessibility of the paper. I think this is important to improve the readability of the current manuscript.

4. What is the consumed time to compose a motion? Will it be too slow for real-time deployment purpose on the robot?

5. No dynamics nor feedback control is discussed / considered.

Overall, this work presents an interesting method to compose motion for legged robots. By addressing the above limitations, future iterations could enhance the paper.

Experimental design

See above.

Validity of the findings

See above.

Additional comments

See above.

·

Basic reporting

English is correct, understandable, without comments.
The topic is relevant, and the literature used is also relevant.
The figures are readable, the notations and abbreviations are clear.
The solution to the proposed hypothesis is present, the article reasonably describes the principles of solutions.
The software is developed and the results obtained are shown in the form of graphs

Experimental design

The research topic fits the Aims and Scope of the journal.
The research objectives are set correctly and lead to the achievement of the research goal.
The research is described in technical language in compliance with ethical standards.
The research methods are described transparently enough for subsequent researchers.

Validity of the findings

Software has been developed that confirms the results of the theoretical studies proposed in this paper.
The data on which the conclusions are based are accepted in specific repositories of the discipline.
The conclusions are substantiated and correspond to the purpose and task of the study.

Additional comments

In general, the article is of great interest. In our opinion, it should be accepted for publication.
Of course, it should be noted that the use of the kinetography method is quite specific in terms of solving the problem of controlling humanoid robots.
However, it is of scientific interest as an alternative method to classical robot control systems.
It should also be noted that the experiment was partially conducted not on a robot, but on a person, who has greater flexibility and smoothness of movements.
However, this is of considerable interest for further research.

Reviewer 3 ·

Basic reporting

See below

Experimental design

See below

Validity of the findings

See below

Additional comments

This research presents a novel approach in developing software for bipedal humanoid robot controllers by constructing and applying graphic domain-specific languages (DSLs) based on kinetography, aiming to enhance the spontaneity and naturalness of robot movements. The study extends DSLs to model dance movements, considering factors like foot topology, sensors, and actuators, and believes this software can be applied to assistive robots for people with mobility challenges. Comments:

The paper is well structured overall, but some parts could be further refined to improve clarity. For example, when introducing DSM and DSLs, you could more clearly distinguish between these concepts and explain how they relate to the development of the robot controller.

The paper should provide more details on the experimental setup, test cases, and results. In particular, quantitative performance metrics and qualitative analysis of the test results of the proposed controller software on a real robot platform should be provided.

What were the criteria you used to validate the effectiveness of your software tool? How comprehensive were your tests in covering various scenarios and edge cases?

This paper lacks a recent related advancements, like Multimodal strain sensing system for shape secognition of tensegrity structures by combining traditional regression and deep learning approaches; and Predictive modeling of flexible EHD pumps using KAN.

Have you considered how users or choreographers might interact with your DSLs? How does your system incorporate user feedback to improve performance?

·

Basic reporting

1. Include a list of nomenclatures at the manuscript's beginning. 
2. Define "labontation" in Section 2.1. 
3. Rename Section 2.3 to "Contributions" to emphasize novelty. 
4. Remove the subtitle from Section 3.5 and integrate it directly. 
5. Change Section 4 to "Methodology." 
6. A comparative study is essential to underscore the contribution.
7. Add a separate section for results and discussions.

Experimental design

The experiments are well-conducted, but a comparative study is needed to emphasize the contribution.

Validity of the findings

The findings should be supported by a comparative study.

Reviewer 5 ·

Basic reporting

See #4 Additional Comments.

Experimental design

See #4 Additional Comments.

Validity of the findings

See #4 Additional Comments.

Additional comments

The authors developed a Domain-Specific Language (DSL) for the movements and actions of bipedal humanoid robots. This is an important topic and interesting work. However, it is not clear what the goal of this paper is and what the research questions are. Moreover, sufficient details on implementation have not been provided. Before acceptance, the presentation must be greatly improved.


1) The authors wrote: "The paper presents a new approach in the development of software for bipedal humanoid robot controllers, based on the construction and application of graphic domain-specific languages (DSLs)." It is my understanding that a new approach is the development and use of a DSL for writing programs for bipedal humanoid robot controllers. However, examples of DSL programs are rare (the authors could provide some examples in the supplement). Furthermore, the application of several graphic domain-specific languages is mentioned, which are hard to identify from the text. How are several DSLs coordinated/orchestrated among themselves?

2) Some sentences are odd and hard to comprehend. For example, "The fact that a finite set of grammar rules can describe an infinite set of sentences - states, is not of great use [3]." It is not clear how this statement is connected with the rest of the statements in the paragraph and what is wrong with grammar when describing the language syntax.

3) The authors wrote: "Each model is a single instance of the language type defined by the meta-model." This wording is not correct. What is a language type in this context? The meta-model describes the structure of models. Hence, a meta-model corresponds to a grammar. On the other hand, a model can be a synonym for a program in model-driven development.

4) The authors wrote: "We cannot reduce the role of the domain expert - Fig. 1(2), to the role of an engineer from a production plant or a researcher from a laboratory. ... They are users as well as participants in the verification of DSL for modeling mechanical properties, movements and actions of robots." This is true for almost every DSL, where the end-users are experts from a domain different from software engineering. References are missing to support such claims. Several works recommend including domain experts (end-users) in DSL design and not only in DSL verification. At this point, it would be beneficial to describe different phases of DSL lifetime (decision, domain analysis, design, implementation, verification, deployment, maintenance). Otherwise, readers who are non-DSL experts will have a hard time comprehending these statements.

5) The authors wrote: "... because it is not enough that only one level in the DSM hierarchy is reliable, but reliability is required at each of the four levels." However, not all four levels in the DSM hierarchy have been introduced previously. It is only later stated what these four levels are (Figure 1(1)). More importantly, it is not clear from the rest of the paper how reliability at all four levels has been increased by the proposed approach.

6) The authors wrote: "Through iterative construction, application and refinement of DSLs, code generators, libraries (framework) and the target system that executes the specifications (e.g. ROS) are developed in parallel." How is this approach different to incremental DSL development?

7) The acronym ROS was not defined in its first usage (page 7).

8) The authors wrote: "This is why our approach relies on software tools that allow fast reference implementation [7] and testing DSLs via model execution [8,5]. Model execution spans all levels in the DSM architecture." However, later in the paper, it is not revealed which software tools have been used to allow fast reference implementation and how testing of the proposed DSL has been realized. References on DSL testing are also missing.

9) The authors wrote: "Finally, in the analyzed papers it is not visible how the models, when they exist, are translated into executable specifications - the controller software, into run-time, for the purposes of updating the control logic. In our opinion, generating good controller software is not possible using only the kinetography language, but that DSL must be extended." However, in this paper, the authors also didn't show how models are translated to executable code, nor did they show how DSL has been extended. It seems that in both cases, an ad-hoc approach was applied.

10) The authors wrote: "The specificity of jobs, which can be performed by bipedal robots, point to the need to develop and apply special "algebra" over numerous different semantic domains of properties." However, this special algebra is not presented in sufficient detail later in the paper.

11) The authors wrote: "Because of all this, we propose a new approach that enables the automatic generation of software for humanoid robot controllers with numerous variations of hardware (feet) and numerous variants of model interpretation." Again, the authors didn't provide sufficient details on how automatic generation from DSL programs onto executable code has been achieved, nor numerous variants of model interpretation.

12) The authors wrote: "Nevertheless, the faithful interpretation is not possible if there are no formally described semantic relations between those wholes." However, it is not clear how the semantics of DSL constructs have been formally defined for the proposed DSL(s).

13) The authors wrote: "... a model query language was created for model searching and updating." However, no specifics about model query language have been provided.

14) Typos:

Roles of experts in robotizatio
-> // Figure 1
Roles of experts in robotization

model interpretation.a.
->
model interpretation.

remoce
->
remove

allebra of curves
->
algebra of curves

involves a fast referent implementation
->
involves a fast reference implementation

---

## Round 0.2 · Minor Revisions

Thank you for your revised submission. However, your latest response did not satisfactorily address the concerns of one of the reviewers. Please carefully review the latest feedback and make a concerted effort to clarify the positioning of your work concerning existing publications, improve the English, and provide more appropriate citations to relevant research.

**Language Note:** The review process has identified that the English language must be improved. PeerJ can provide language editing services - please contact us at [email protected] for pricing (be sure to provide your manuscript number and title). Alternatively, you should make your own arrangements to improve the language quality and provide details in your response letter. – PeerJ Staff

Reviewer 5 ·

Basic reporting

The authors haven't revised the paper to my expectations. Several comments have been omitted and have not been addressed in the revised manuscript. In particular (I am using previous numbering):

2) Some sentences are odd and hard to comprehend. For example, "The fact that a finite set of grammar rules can describe an infinite set of sentences - states, is not of great use [3]."
What do the authors want to express with this statement? What is wrong with using grammar to describe the language syntax?

4a) … This is true for almost every DSL, where the end-users are experts from a domain different from software engineering. References are missing to support such claims. Several works recommend including domain experts (end-users) in DSL design and not only in DSL verification.

4b) At this point, it would be beneficial to describe different phases of DSL lifetime (decision, domain analysis, design, implementation, verification, deployment, maintenance).

6) The authors wrote: "Through iterative construction, application and refinement of DSLs, code generators, libraries (framework) and the target system that executes the specifications (e.g. ROS) are developed in parallel." How is this approach different to incremental DSL development?
References for incremental DSL development are missing.

8) References on DSL testing are also missing.

12) The authors wrote: "Nevertheless, the faithful interpretation is not possible if there are no formally described semantic relations between those wholes." However, it is not clear how the semantics of DSL constructs have been formally defined for the proposed DSL(s).
In the revised paper, the authors now claimed that semantics is given by meta-models ("The semantics of each DSL used for these purposes, as well as the semantics of each graphical DSL we used, is defined by means of a meta-model."). That claim is wrong since meta-models define only the structure of the DSL (its syntax). Semantics are defined with other methods, such as operational semantics, denotational semantics, and translational semantics. Without proper semantic description, the authors can't claim that "The definition of DSL is therefore completely formal, and every language concept has at least one visual representation." Namely, visual representation does not define semantics.

Experimental design

See comments above.

Validity of the findings

See comments above.

---

## Round 0.3 · accepted · Accept

Upon carefully reviewing the authors' response to the feedback provided in the previous round, I am pleased to confirm that the manuscript has addressed all outstanding issues and concerns. I am satisfied with the manner in which the authors have resolved the points raised.